# The Adsorption Behavior of Moisture on Smelter Grade Alumina during Transportation and Storage—for Primary Aluminum Production

**Youjian Yang** [1,2,*] **, Wenju Tao** [1,2] **, Weicheng Liu** [1,2] **, Xianwei Hu** [1,2] **, Zhaowen Wang** [1,2] **, Zhongning Shi** [1,2] **and Xin Shu** [3]

1    School of Metallurgy, Northeastern University, Shenyang 110819, China; taowj@smm.neu.edu.cn (W.T.); yangyou100@163.com (W.L.); huxw@smm.neu.edu.cn (X.H.); wangzw@smm.neu.edu.cn (Z.W.); shizn@smm.neu.edu.cn (Z.S.)
2    Key Laboratory for Ecological Metallurgy of Multimetallic Mineral (Northeastern University), Ministry of Education, Shenyang 110819, China
3    Hunan Aerospace TianLu Advanced Material Testing Co., Ltd. Changsha 410600, China; shuxshux@163.com
*    Correspondence: yangyj@smm.neu.edu.cn; Tel.:+86-24-8368-6464

**Abstract:** Smelter grade alumina (SGA) plays multiple roles in the Hall–Héroult process for primary aluminum production. Given its very porous nature, one major role of SGA is to adsorb toxic hydrogen fluoride (HF) in the dry scrubber. However, also because of its porous nature, SGA inevitably adsorbs ambient moisture. This paper discusses the influence of alumina properties, including pore size distribution and specific surface area, on the physical adsorption of water vapor on SGA, as well as the adsorption kinetics. The result shows that the adsorption enthalpy of moisture on SGA is in the range of 4–13 kJ/mol. The adsorption capacity increases significantly with the particle specific surface area and total pore volume. A higher adsorption temperature indicates a much faster adsorption rate but corresponds to a lower equilibrium adsorption capacity.

**Keywords:** alumina; physical adsorption; moisture adsorption; aluminum electrolysis

## 1. Introduction

The smelter grade alumina (SGA) for primary aluminum production is produced with the Bayer process. During the Bayer alumina production process, the SGA is obtained from the calcination of gibbsite under the temperature of 1000–1200 °C depending on different calciner technologies. The product SGA usually has a purity of more than 98.5%, and the phase composition of SGA generally consists of 2–10% $\alpha$-Al$_2$O$_3$, 25–40% $\theta$-Al$_2$O$_3$, 50–75% $\gamma/\gamma'$-Al$_2$O$_3$, and a little bit of boehmite (less than 1%). Due to the very porous nature of $\gamma/\gamma'$-Al$_2$O$_3$ phases, the SGA behaves with a large Brunauer, Emmett and Teller (B.E.T.) surface area of 50–120 m$^2$/g [1]. This large specific surface area substantially benefits the dissolution of SGA in the molten electrolyte and the dry scrubbing capacity during the Hall–Héroult aluminum production process.

During the calcination of SGA from gibbsite in the Bayer process, some structural hydroxyl remains in the product SGA to obtain optimized physicochemical properties of the product SGA. The structural hydroxyl is mainly contained in the boehmite and $\gamma/\gamma'$-Al$_2$O$_3$ phases, where the boehmite phase decomposes at 500–550 °C [1]. The structural hydroxyl content of SGA is usually measured with the loss on ignition (LOI 300–1000) between 300 °C and 1000 °C, and the reasonable LOI value of SGA is approximately 0.3–1.5%. The hydroxyl content in SGA contributes to nearly 25–40% of the total hydrogen fluoride (HF) emission in the aluminum reduction cell by the occurring hydrolysis reaction with fluoride electrolyte [2].

Owing to the porous nature of SGA particles, SGA absorbs moisture from the atmosphere during transportation and storage. The physisorbed moisture, called the moisture content of SGA, can be easily removed by drying at 110 °C for 24 h, and is usually quantitatively described with the value of moisture on ignition (MOI 25–300) between 25 °C and 300 °C. The physisorbed moisture influences the industrial aluminum electrolysis process in several respects. The SGA moisture is introduced to the molten fluoride electrolyte through alumina feeding and causes fluoride hydrolysis, contributing to nearly 10–25% of the total HF emission in the original cell fume [3,4]. Most of the physisorbed moisture is flashed off before reaching the high-temperature melt surface, increasing the relative humidity (RH) of the cell exhaust gas. The exhaust gas containing moisture and HF is then fully adsorbed by SGA in the gas treatment center (GTC), increasing the physisorbed moisture and fluorine content of the secondary SGA. The dry scrubber is designed to recycle the solid particles and gaseous HF, but evidence exists that both the high humidity in the gas and high physisorbed moisture content of the SGA benefit HF adsorption [5].

Ambient humidity also significantly influences the SGA physisorbed moisture content [6]. The reasonable range for SGA MOI value is 1.5–4%, though the MOI value can exceed 4–5% for plants located in hot and humid areas or during summer. Some earlier studies have also mentioned that a high MOI value facilitates alumina dissolution because the flashed-off water vapor disperses the alumina particles upon feeding [7].

The physisorbed moisture on SGA results in extra HF emissions through feeding, increasing the adsorption burden for SGA in the dry scrubber. This appears to be contradictory because, owing to its large specific surface area, SGA exhibits good adsorbability for both HF and water vapor. However, this fact can help reveal the HF adsorption process on SGA. Before an HF molecule is chemically adsorbed onto the alumina surface, it first undergoes a physical adsorption process that resembles that of a water molecule on the alumina surface.

The adsorption of moisture and HF onto SGA are closely related. HF adsorption onto alumina particles has been demonstrated to be a chemisorption process where the fluorine atom is chemically bonded to the aluminum atom, and both the diffusion and reaction rates can affect the adsorption rate [8]. As a preceding physical process, the diffusion rates of gaseous moisture and HF into the internal surface are expected to be very similar. Meanwhile, the increased humidity at the active site accelerates the chemisorption rate of HF.

There are two motivations for this work. First, this work introduces the physical adsorption capacity of alumina. Since the adsorption of moisture and HF on SGA are closely related, the investigation on the diffusion rates of gaseous moisture can have some indications on the adsorption behavior of HF on the SGA. Second, as the SGA adsorbs ambient moisture during storage and transportation, the increased water content in SGA would result in an increase in the cell HF emission after alumina feeding due to electrolyte hydrolysis. This investigation could give some basic data for a better understanding of the cell 'summer syndrome'.

## 2. Experimental

### 2.1. Experimental Design and Sample Preparation

In this work, three alumina samples marked #1, #2, #3 were tested to determine their specific surface area, particle size distribution, and pore size distribution, respectively. Sample #1 was sampled from a Chinese aluminum smelter, sample #2 was extracted from an Australian aluminum smelter, and sample #3 was manufactured from coal fly ash where hydrochloric acid was used as the leaching agent for fly ash raw material. The adsorption capacities of water vapor on these three alumina samples were then compared. Then, sample #1 was pre-calcined at 800 °C, 1000 °C, and 1200 °C for 2 h, respectively, and adsorption tests were subsequently performed on calcined samples. The adsorption kinetics were analyzed to reveal the adsorption behavior of gaseous molecules on the SGA surface.

Referring to the heat treatment carried out on the Bayer SGA sample #1, the calcination process changed the microstructure of the SGA particles. The dominant phase of SGA changed from $\gamma$-$Al_2O_3$ to $\alpha$-$Al_2O_3$ after the calcination at 1200 °C for 2 h. The phase transformation caused a significant decrease in the specific surface area and pore volume. The measurement data of the adsorption capacity before and after calcination could provide evidence for the relationship between the moisture adsorption capacity and specific surface area/pore volume.

China is facing a problem of shortage in high-quality bauxite now. Some alumina companies are trying to find a new method to produce smelter grade alumina from high-alumina-content coal fly ash. Since the Al/Si ratio in coal fly ash is approximately 1, the Bayer process is not suitable. Some companies have made their first step successfully. Sample #3 tested in this work is a kind of alumina produced from coal fly ash. It is quite a special contrast sample to the Bayer alumina, and some comparisons have been made in this work to show some first-hand data.

The experimental design is shown in Table 1.

**Table 1.** Sample treatment and experimental design.

| Sample | Pre-Treatment | Tests Carried Out | Remarks |
|---|---|---|---|
| #1 | 300 °C, 6 h | Water vapor adsorption tests at 20 °C, 40 °C, and 60 °C, respectively; Brunauer, Emmett and Teller (B.E.T.) surface area, Pore size distribution, Particle size distribution, Loss on ignition (LOI 300–1000). | Sampled from a Chinese aluminum smelter |
| #2 | 300 °C, 6 h | | Sampled from an Australian aluminum smelter |
| #3 | 300 °C, 6 h | | Manufactured from coal fly ash |
| #1 | 800 °C, 2 h | Water vapor adsorption tests at 20 °C; B.E.T. surface area, Pore size distribution. | Pre-calcined Chinese alumina, Sample #1 |
| | 1000 °C, 2 h | | |
| | 1200 °C, 2 h | | |

## 2.2. Water Vapor Adsorption Tests

The water vapor adsorption tests were performed on a water sorption analyzer, manufactured by Quantachrome Instruments (Model: Aquadyne DVS, Graz, Austria). With this equipment, the water adsorption curve of an alumina sample could be measured using a microbalance in response to changes in the partial steam pressure.

Before the water adsorption test, 20–25 mg of alumina sample (dried at 300 °C for 6 h before testing) was positioned in one of the two pans of a microbalance (accuracy of 0.1 µg). First, dry nitrogen was injected to purge the adsorption chamber of excess moisture, and the temperature was stabilized at the preset temperature. When the test began, the relative humidity (RH) in the chamber was increased according to the program by adjusting the proportion of water vapor and dry nitrogen. During the test, the RH in the chamber was programmed to increase by 10% increments from 0% to 90% and then decrease by the same increments to 0%. At each step, sufficient adsorption time was allowed for the sample to reach equilibrium. The adsorption capacity at each step was obtained from the weight change of the alumina sample measured by the microbalance.

The B.E.T. surface area and pore size distribution were measured using the nitrogen adsorption method with a 3H-2000PW analyzer (manufactured by Beishide Instrument, Beijing, China). The particle size distribution was determined with a laser particle size analyzer manufactured by Malvern Instruments (Mastersizer 2000, London, UK). The MOI value was calculated as the percentage weight loss between 25 °C and 300 °C over 1h. The LOI value was calculated as the percentage weight loss between 300 °C and 1000 °C over 1 h.

## 3. Results and Discussion

### 3.1. Properties of Tested Alumina Samples

Several adsorption-related properties of alumina samples #1, #2, and #3 are listed in Table 2.

**Table 2.** Several adsorption-related properties of the tested alumina samples.

| Testing Item | Parameters | Sample #1 | Sample #2 | Sample #3 |
|---|---|---|---|---|
| Specific surface area | B.E.T. surface area/m$^2$ g$^{-1}$ | 68.6 | 74.0 | 91.5 |
| | D(10)/μm | 46.2 | 32.4 | 65.1 |
| Particle size distribution | D(50)/μm | 85.5 | 67.2 | 167 |
| | D(90)/μm | 140 | 121 | 365 |
| | Mean pore size/nm | 8.70 | 8.69 | 13.31 |
| Pore size distribution | Most probable pore size/nm | 7.68 | 7.41 | 11.12 |
| | Total pore volume/mL g$^{-1}$ | 0.2426 | 0.2683 | 0.4107 |
| Hydroxyl content | LOI (300–1000)/% | 0.56 | 0.66 | 2.35 |

In Table 2, the D(10), D(50), and D(90) values indicate the critical diameters below which the small particles account for 10%, 50%, and 90% of the total particle volume, respectively, of the total particles. The D(50) value is usually regarded as the average particle size. The mean pore size is derived from the cylinder model in which the actual irregularly shaped pores are assumed to be standard cylinders, and the value is calculated as four times the total pore volume divided by the specific surface area. The most probable pore size is read from the location of the strongest peak on the pore size distribution curve. For a sample in which the pore size is normally distributed, the mean pore size should equal the most probable pore size.

Sample #3 had the highest B.E.T. surface area of the three samples, containing coarser particles, larger pores, and a wider pore size distribution than those of the other two samples. Notably, the LOI (300–1000) of sample #3 was as high as 2.35%, which contributed significantly to HF emission from the aluminum reduction cell [9].

The particle size distribution curves of samples #1, #2, and #3 are shown in Figure 1. From a particle size perspective, samples #1 and #2 appear to be more appropriate for industrial operations because they exhibited a more uniform particle size (narrower particle size range than sample #3, in Figure 1), fewer fine particles, and fewer large particles than sample #3 did.

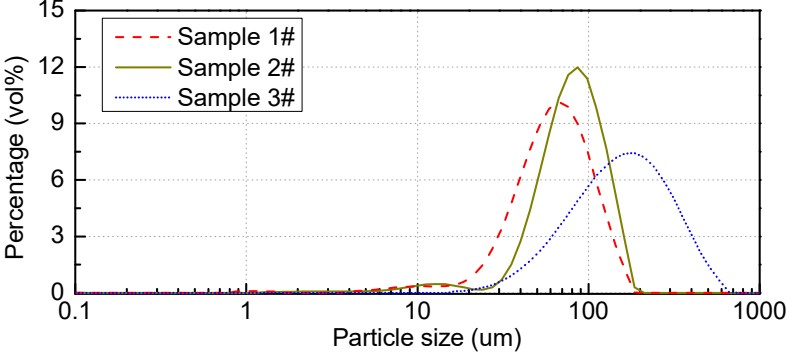

**Figure 1.** The particle size distribution curves of samples #1, #2, and #3.

The pore size distributions of samples #1, #2, and #3 are shown in Figure 2. The accumulated pore volume curve displays the total pore volume of all the micropores. For the Bayer alumina, the total pore volume of sample #2 (0.2444 mL/g, Figure 2b) slightly exceeded that of sample #1 (0.2206 mL/g,

Figure 2a), with this higher pore volume corresponding to a larger B.E.T. surface area (74.0 m$^2$/g for sample #2 versus 68.6 m$^2$/g for sample #1, Table 2). Finally, it is easy to conclude that sample #2 presented a more porous microstructure than that of sample #1.

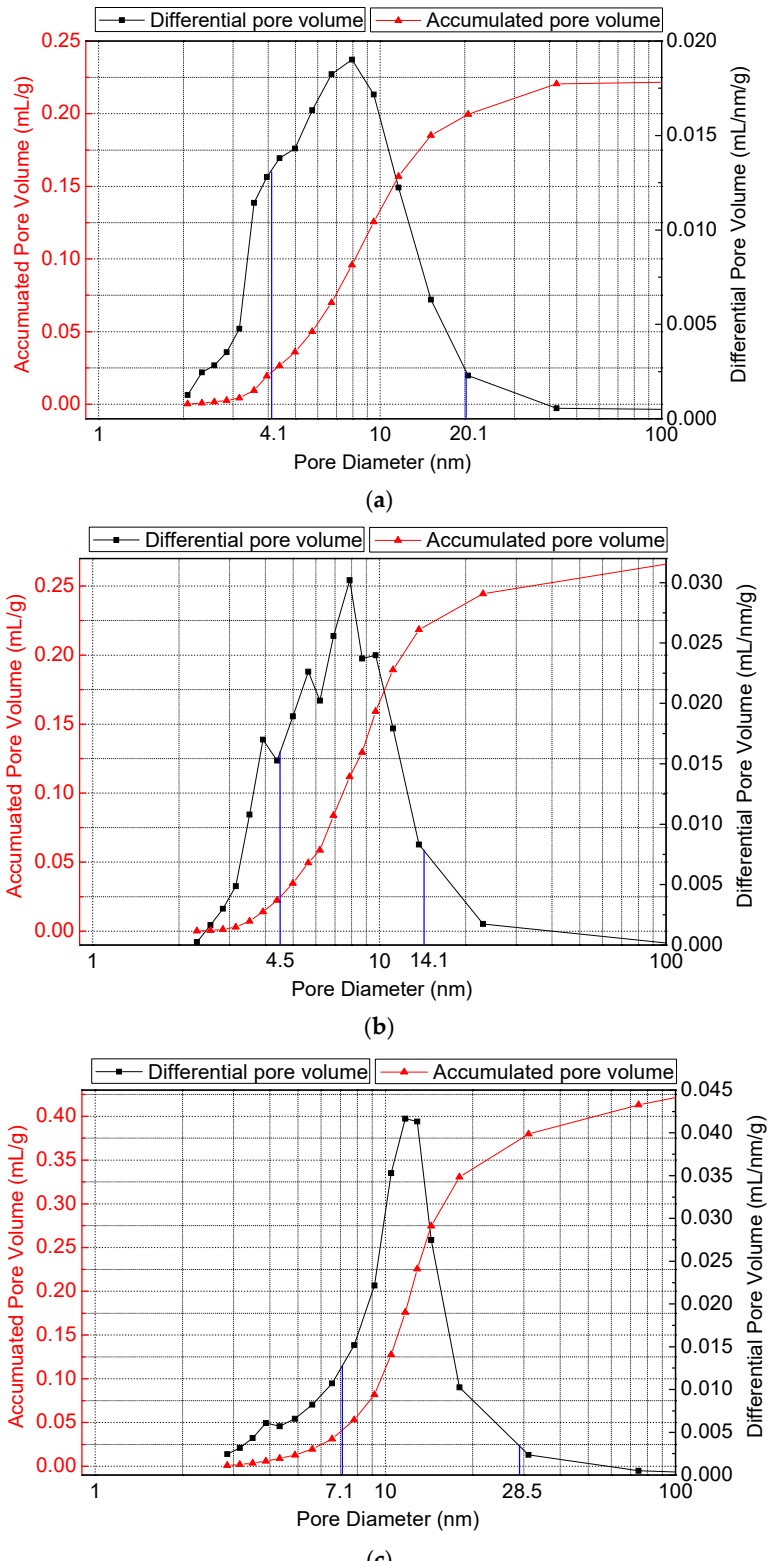

**Figure 2.** The pore size distribution curves of samples #1, #2, and #3; (**a**) sample #1, (**b**) sample #2, (**c**) sample #3.

However, the coal-fly-ash-manufactured alumina—sample #3—presented the largest total pore volume and B.E.T. surface area of the three samples at 0.4131 mL/g (Figure 2c) and 91.5 m²/g (Table 2), respectively. A large micropore volume and specific surface area empirically indicate good SGA adsorption capacity for gaseous HF and particulates.

In Figure 2, the pore-concentrated region for each sample is marked. This pore-concentrated region is defined as the pore size interval from P(10) to P(90), where P(10) is defined as the pore size below which the accumulated pore volume comprises 10 vol% of the total pore volume, and P(90) is the pore size below which the accumulated pore volume comprises 90 vol% of the total pore volume. The pore concentration regions of the three samples are compared in Table 3.

**Table 3.** A comparison of the pore concentration region between the three alumina samples.

| Sample | Total Pore Volume/mL g⁻¹ | B.E.T. SA/m² g⁻¹ | P(10)/nm* | P(90)/nm* |
|:---:|:---:|:---:|:---:|:---:|
| #1-Bayer | 0.2206 | 68.6 | 4.1 | 20.1 |
| #2-Bayer | 0.2444 | 74.0 | 4.5 | 14.1 |
| #3 | 0.4131 | 91.5 | 7.1 | 28.5 |

\* P(10) and P(90) are the pore sizes below which the accumulated pore volume comprises 10 vol% and 90 vol% of the total pore volume, respectively.

From Table 3, the pore size of sample #1 was concentrated between 4.1 and 20.1 nm, while that of sample #2 was concentrated in a narrower distribution range between 4.5 and 14.1 nm. Combining this with the analysis of the particle size measurement, in which sample #2 also exhibited a narrower particle size range than sample #1 (Figure 1), it is reasonable to conclude that sample #2 had more uniform particle properties than the other two samples.

### 3.2. Water Vapor Adsorption on Fresh SGA

The water vapor adsorption tests were carried out at 20 °C, 40 °C, and 60 °C. The adsorption isotherms of water vapor on the three SGA samples are shown in Figure 3, which shows that the adsorption isotherms coincided well with the classical *IUPAC* (International Union of Pure and Applied Chemistry [10]) *Type V* isotherm, suggesting a weak interaction between the adsorbent and adsorbate. Under all circumstances, a distinct adsorption/desorption hysteresis loop was observed, and the shape of the hysteresis loop closely matched the *H3 type*, in which the loop extended to the high RH range where no limiting adsorption was observed. The *H3* hysteresis loop type also indicates that the most likely mesopore structure of the porous adsorbent comprised slit-shaped pores. While the hysteresis loop did not extend to the low RH area, it does indicate pure physical adsorption of water vapor on the SGA surface rather than chemical adsorption.

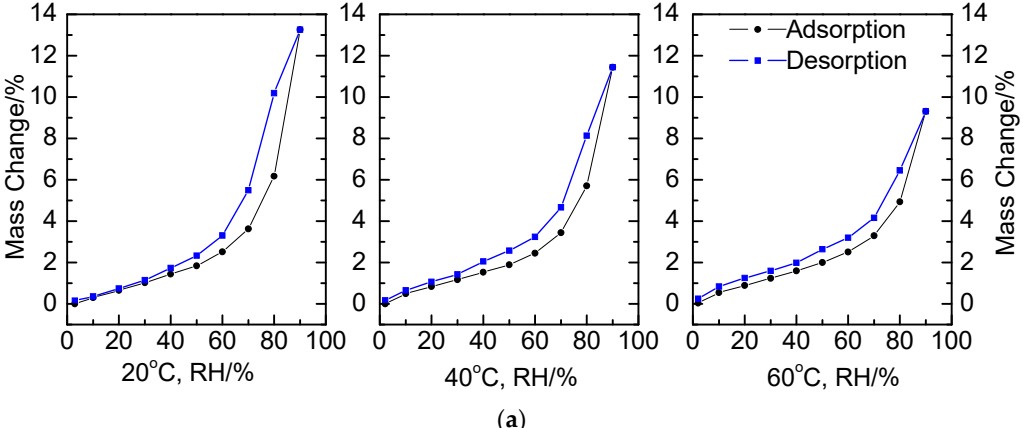

**Figure 3.** *Cont.*

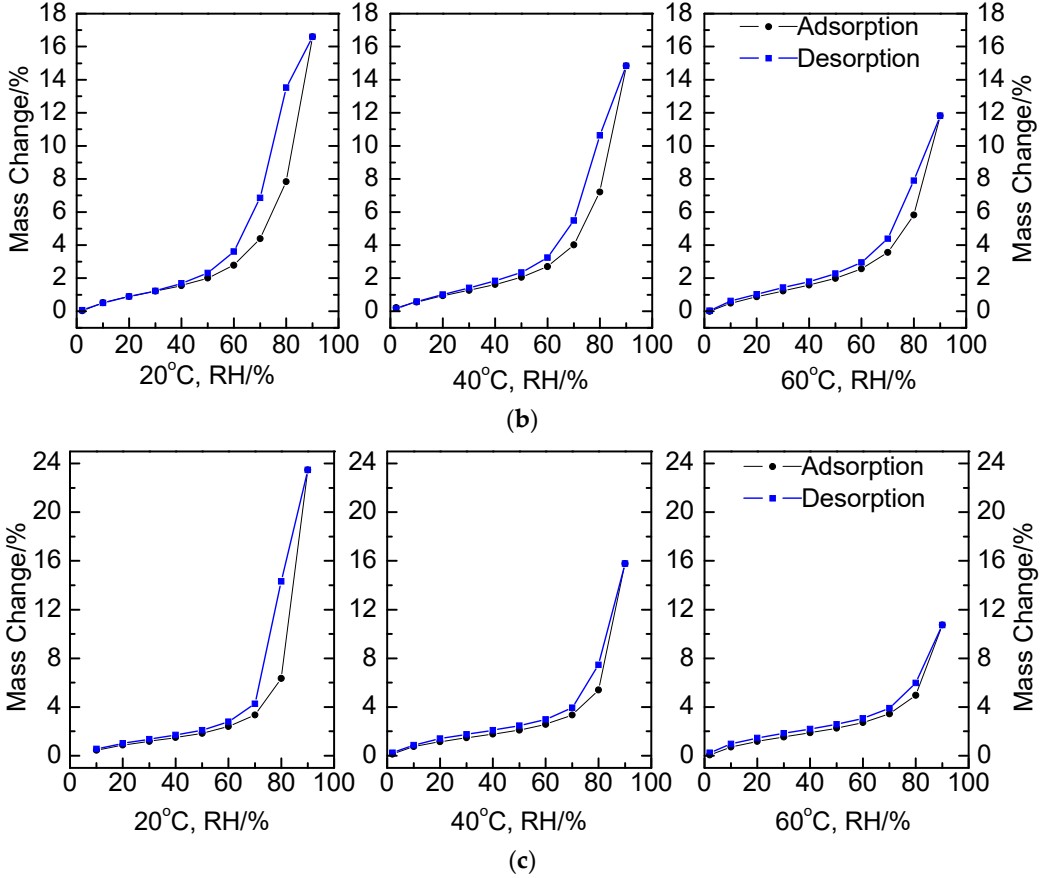

**Figure 3.** Water vapor adsorption/desorption isotherms on SGA (smelter grade alumina) at 20 °C, 40 °C, and 60 °C. (**a**) sample #1, (**b**) sample #2, (**c**) sample #3.

Considering that an entire adsorption–desorption loop was observed, the adsorption of moisture on SGA was determined to be a completely reversible process. For physisorption, the equilibrium is highly dependent on temperature, whereby low temperature is usually conducive to the adsorption process, while high temperature accelerates desorption. Therefore, the equilibrium adsorption capacities for all RH decreased with increasing temperature (Figure 3).

The equilibrium adsorption capacities at RH 90% for the three samples under different temperatures are compared, see Figure 3 or Table 4. For each SGA sample, the moisture content of SGA was significantly influenced by the ambient temperature and RH. Normally, SGA contains 2–4% of moisture during the smelter operation, and this amount of adsorbed water has been reported to significantly contribute to HF formation and emission [11,12]. The value of equilibrium capacity shows the potential adsorptive property of ambient moisture for the specific SGA batch.

**Table 4.** The equilibrium adsorption capacities at relative humidity (RH) 90% of samples #1, #2, and #3.

| Temperature/°C | Max Equilibrium Adsorption Capacity | | |
| --- | --- | --- | --- |
| | Sample #1 | Sample #2 | Sample #3 |
| 20 | 7.3664 | 9.2188 | 13.0421 |
| 40 | 6.3536 | 8.2413 | 8.7694 |
| 60 | 5.1674 | 6.5675 | 5.9639 |

The adsorption capacities differed among the three SGA samples tested in this work. The essential cause of these differences was the micropore morphology of the SGA. Between the two Bayer SGAs,

sample #2 exhibited a faster adsorption rate and larger adsorption capacity for water vapor compared to sample #1. However, the chemical adsorption of gaseous HF on SGA is believed to be more complicated based on a subsequent chemisorption process after the diffusion of HF molecules into the micropores of the SGA particle. Many earlier studies have investigated the kinetics of HF adsorption on SGA, demonstrating that the HF adsorption rate is highly related to the moisture content [6], sodium content [13], specific surface area, and pore size distribution [14]; however, the adsorption mechanism of HF remains unclear [8]. Nevertheless, given the similar diffusion dynamics of gas molecules in mesoporous materials, some initial adsorption processes of HF may resemble the mass transfer process of $H_2O$ from the outside to the internal pore surface of SGA.

### 3.3. Water Vapor Adsorption on Calcined SGA

Here, the SGA sample #1 was pre-calcined at 800 °C, 1000 °C, and 1200 °C for 2 h, respectively. The water vapor adsorption tests were performed at 20 °C on the calcined samples to investigate the effects of phase transition on the SGA adsorption capacity. Table 5 shows the changes in the B.E.T. surface area and several pore size parameters after calcination, and Figure 4 shows the pore size distribution curves of pre-calcined sample #1. The B.E.T. surface area and total pore volume clearly and significantly decreased with increasing calcination temperature. During calcination, the dominant phase of SGA gradually changed from the gamma phase to the alpha phase. After the 2-h pre-calcination at 1200 °C, the alpha phase content of the SGA sample accounted for more than 90% (alpha phase content of fresh SGA was less than 10%). The phase transition rate was high from 1000 °C to 1200 °C, corresponding to a rapid reduction in the B.E.T. surface area and total pore volume. However, parameters, such as mean pore size and most probable pore size, showed no distinct pattern.

**Table 5.** B.E.T. (Brunauer, Emmett, and Teller) surface area and pore size parameters of calcined sample #1.

| Treatment Condition | B.E.T. SA /m²·g⁻¹ | Total Pore Volume/mL g⁻¹ | Mean Pore Size/nm | Most Probable Pore Size/nm |
|---|---|---|---|---|
| 300 °C, 6 h | 68.6 | 0.2426 | 8.70 | 7.68 |
| 800 °C, 2 h | 61.3 | 0.2184 | 8.75 | 7.30 |
| 1000 °C, 2 h | 40.9 | 0.2089 | 14.24 | 11.20 |
| 1200 °C, 2 h | 11.7 | 0.0533 | 12.41 | 2.47 |

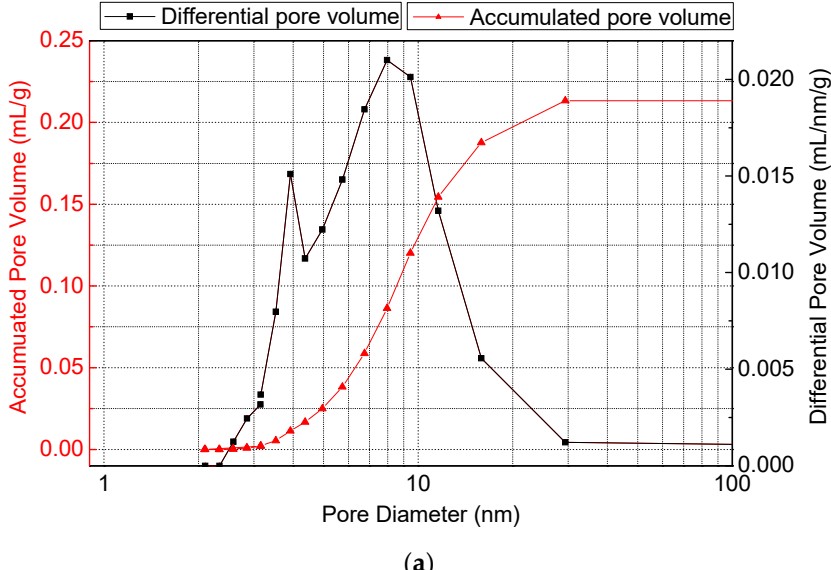

(**a**)

**Figure 4.** *Cont.*

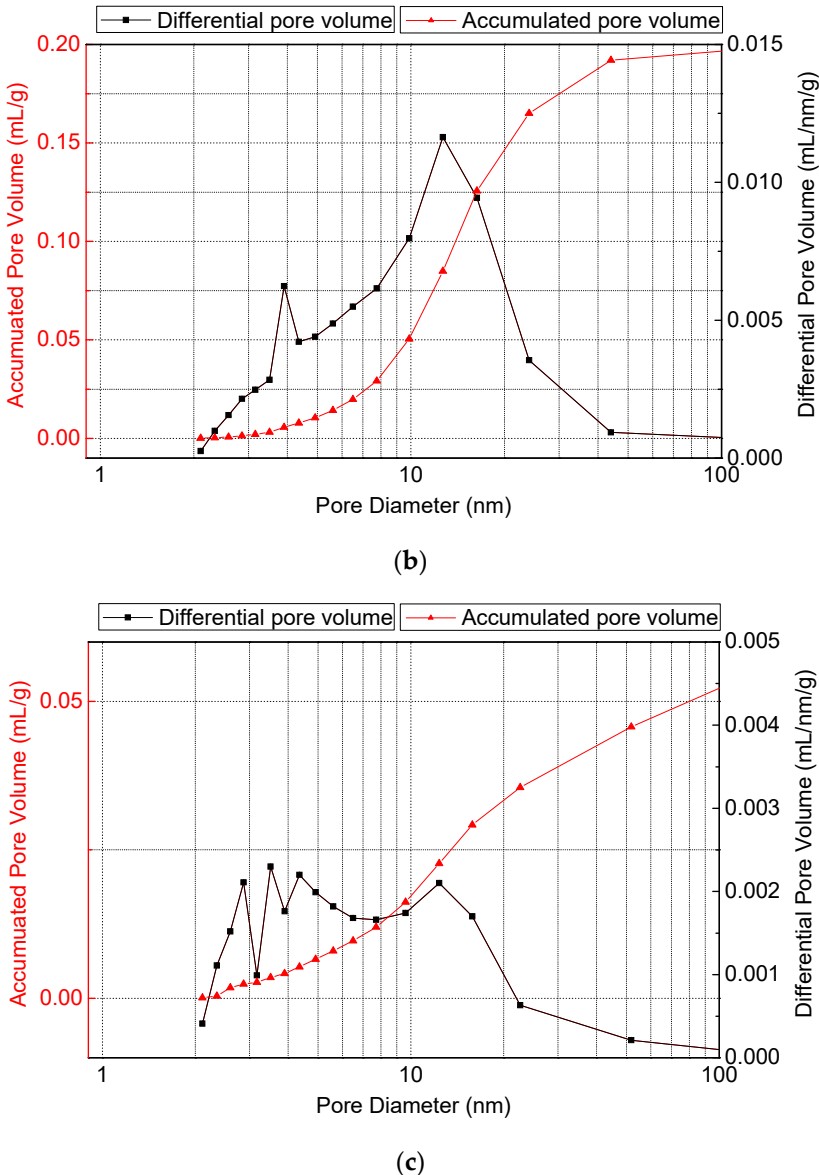

**Figure 4.** The pore-size distribution curves of pre-calcined sample #1 at 800 °C, 1000 °C, and 1200 °C for 2 h. (**a**) 800 °C, 2 h (**b**) 1000 °C, 2 h (**c**) 1200 °C, 2 h.

The water vapor adsorption isotherms of the calcined samples are shown in Figure 5. A comparison of the isotherms of the pre-calcined SGAs showed that the adsorption capacity decreased significantly with increasing calcination temperature, especially from 1000 °C to 1200 °C, which also corresponded to the large reduction in B.E.T. surface area and total pore volume.

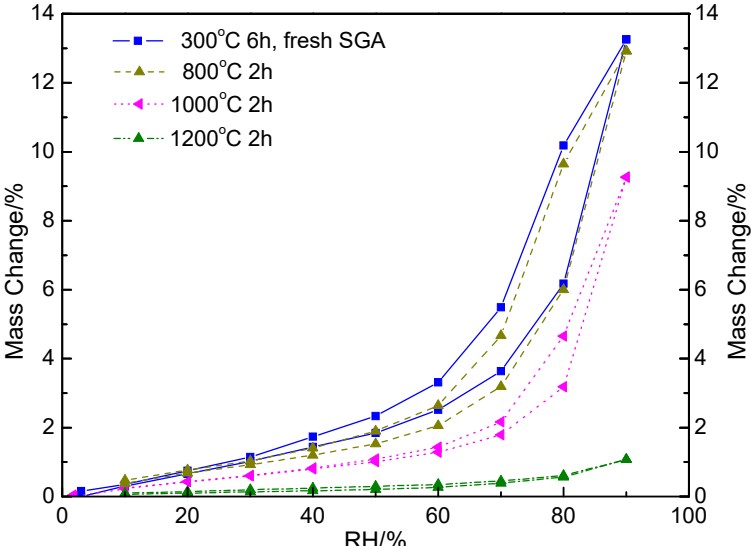

**Figure 5.** The water vapor adsorption isotherms of the calcined sample #1 at 800 °C, 1000 °C, and 1200 °C for 2 h.

### 3.4. Calculation of the Adsorption Enthalpy

The B.E.T. equation (Equation (1)) was used to calculate the adsorption enthalpy of moisture on SGA. The basic hypothesis of the B.E.T. model is that multi-layer gas adsorption occurs on a uniform solid surface. With the B.E.T. model, parameters $V_m$ and $c$ can be calculated from the slope and intercept of the linear fitting curve between $\frac{p}{V(p_s-p)}$ and $\frac{p}{p_s}$, as shown in Equation (2). Constant $c$ is related to the adsorption enthalpy $E_1$ (Equation (3)), which essentially determines the shape of the adsorption isotherm.

B.E.T. equation:

$$\frac{V}{V_m} = \frac{cp}{(p_s-p)[1+(c-1)p/p_s]} \tag{1}$$

Can be expressed as:

$$\frac{p}{V(p_s-p)} = \frac{1}{cV_m} + \frac{(c-1)p}{cV_mp_s} \tag{2}$$

$c$ is related to the adsorption enthalpy:

$$c = \exp\left(\frac{E_1}{RT}\right), (c > 2) \tag{3}$$

$V$ is the adsorption volume, $V_m$ is the single-layer adsorption volume, $p$ is pressure, $c$ is a constant related to the adsorption enthalpy, $p_s$ is the saturated water vapor pressure at corresponding temperature, $E_1$ is the adsorption enthalpy, $R$ is the ideal gas constant, and $T$ is temperature.

The calculated adsorption enthalpy is shown in Table 6. The adsorption enthalpy was as low as 4–13 kJ/mol, which confirmed a physical adsorption mechanism rather than a chemical process. The $R^2$ value was near 1, indicating good agreement between the adsorption results and the B.E.T. equation.

**Table 6.** The adsorption enthalpy of moisture on smelter grade alumina (SGA) derived from the B.E.T. model.

| Testing Sample | Adsorption Temperature/°C | Adsorption Enthalpy/kJ mol$^{-1}$ | Linear Fitting $r^2$ |
|---|---|---|---|
| Sample #1 | 20 | 4.43 | 0.965 |
|  | 40 | 6.68 | 0.991 |
|  | 60 | 9.49 | 0.980 |
| Sample #1, 800 °C, 2 h |  | 5.96 | 0.970 |
| Sample #1, 1000 °C, 2 h | 20 | 6.64 | 0.996 |
| Sample #1, 1200 °C, 2 h |  | 8.10 | 0.991 |
| Sample #2 | 20 | 6.45 | 0.993 |
|  | 40 | 6.87 | 0.989 |
|  | 60 | 6.96 | 0.994 |
| Sample #3 | 20 | 7.34 | 0.999 |
|  | 40 | 10.59 | 0.991 |
|  | 60 | 13.01 | 0.982 |

### 3.5. Modeling the Adsorption Kinetics of Water Vapor on SGA

A comparison of different adsorption kinetic models revealed that the pseudo-second-order rate model matches the testing results better than do the pseudo-first-order rate and in-particle diffusion models. The second-order kinetic equation (Equation (4)) is based on an adsorption-controlling process, in which the adsorption rate is determined by the number of unoccupied active sites.

Pseudo-second-order kinetic equation:

$$\frac{d_q}{d_t} = k_2 (q_e - q)^2 \tag{4}$$

Equation (4) can be expressed as:

$$\frac{t}{q} = \frac{1}{k_2 q_e^2} + \frac{t}{q_e} \tag{5}$$

where $q$ is the adsorption capacity at adsorption time $t$, $k_2$ is the adsorption rate constant, and $q_e$ is the equilibrium adsorption capacity.

In the pseudo-second-order kinetic model, parameters $k_2$ and $q_e$ can be derived from Equation (5), where $\frac{t}{q}$ and $\frac{1}{q_e}$ show a linear relationship. The calculated results of the equilibrium adsorption capacity, *calculated* $q_e$, and the tested saturated adsorption capacity, *tested* $q_e$, are compared in Table 7.

From Table 7, the adsorption rate constant $k_2$ increased with increasing adsorption temperature, which implies that the adsorption rate of moisture on SGA increases at higher ambient temperatures. The $k_2$ value decreased with increasing RH, in which a higher RH corresponds to a larger equilibrium capacity. This indicates that more time is needed to reach a larger equilibrium capacity under higher RH conditions, which benefits the adsorption process. The *calculated* $q_e$ value was very close to the *tested* $q_e$ value, which further confirms that the actual adsorption process agrees well with the pseudo-second-order kinetic model.

Referring to the pre-calcined SGA samples, as shown in Table 7, the adsorption rate under the same RH increased significantly with calcination temperature; meanwhile, the equilibrium adsorption capacity simultaneously decreased significantly. This phenomenon suggests that a low equilibrium is easy to achieve rapidly, suggesting that the nanoscale micropore structure of the SGA is seriously damaged after the phase transition to the alpha phase during pre-calcination.

**Table 7.** Comparison of the calculated (*Calculated $q_e$*) and tested (*Tested $q_e$*) equilibrium adsorption capacities from the pseudo-second-order kinetic model.

| Sample #1 | | | | | |
|---|---|---|---|---|---|
| Sample and Adsorption Temperature | RH /% | $k_2$ | Calculated $q_e$ /mg·g$^{-1}$ | Tested $q_e$ /mg·g$^{-1}$ | Linear $r^2$ |
| Sample #1, 20 °C | 70 | 0.0171 | 37.31 | 36.32 | 0.999 |
| | 80 | $2.863 \times 10^{-3}$ | 65.79 | 61.76 | 0.996 |
| | 90 | $5.777 \times 10^{-4}$ | 138.89 | 132.53 | 0.998 |
| Sample #1, 40 °C | 70 | 0.0683 | 34.45 | 34.36 | 0.999 |
| | 80 | 0.0103 | 57.64 | 57.04 | 0.999 |
| | 90 | $1.583 \times 10^{-3}$ | 117.37 | 114.36 | 0.999 |
| Sample #1, 60 °C | 70 | 0.198 | 32.99 | 32.97 | 0.999 |
| | 80 | 0.0873 | 49.36 | 49.29 | 0.999 |
| | 90 | 0.0175 | 93.28 | 93.01 | 0.999 |
| 800 °C 2 h calcined sample #1, 20 °C | 70 | 0.0119 | 32.82 | 31.90 | 0.999 |
| | 80 | $1.858 \times 10^{-3}$ | 63.25 | 59.99 | 0.998 |
| | 90 | $3.384 \times 10^{-4}$ | 138.89 | 128.85 | 0.998 |
| 1000 °C 2 h calcined sample #1, 20 °C | 70 | 0.1317 | 18.02 | 17.94 | 0.999 |
| | 80 | $7.994 \times 10^{-3}$ | 32.82 | 31.84 | 0.999 |
| | 90 | $3.156 \times 10^{-4}$ | 101.52 | 92.58 | 0.996 |
| 1200 °C 2 h calcined sample #1, 20 °C | 70 | 0.5887 | 3.85 | 3.88 | 0.999 |
| | 80 | 0.1751 | 5.78 | 5.71 | 0.999 |
| | 90 | 0.0419 | 10.74 | 10.78 | 0.999 |

## 4. Conclusions

Based on the investigations of moisture adsorption onto SGA, the following conclusions are reached:

The adsorption of moisture onto SGA is purely a physical process, as proposed by earlier researchers (adsorption enthalpy 4–13 kJ/mol), and the adsorption capacity is significantly related to the B.E.T. surface area and total pore volume, but not to the mean pore size or most probable pore size. A higher adsorption temperature indicates a much faster adsorption rate but corresponds to a lower equilibrium adsorption capacity. The internal microstructure of SGA particles changes markedly after the phase transition to the alpha phase, exhibiting a significant reduction in the B.E.T. surface area and total pore volume, as well as the equilibrium capacity. The modeling analysis showed that the adsorption process closely matches the classical pseudo-second-order kinetic model, which indicates the involvement of an adsorption-controlling process. With this model, the adsorption rate constant $k_2$ and equilibrium adsorption capacity $q_e$ were calculated. The results showed that a higher adsorption temperature benefits the adsorption rate but corresponds to a lower equilibrium capacity.

**Author Contributions:** For the preparation of this article, Y.Y. supervised the experiments, reviewed and edited the manuscript; W.T. helped with the methodology; W.L. conducted most of the experiments; X.H., Z.W., and Z.S. offered part of the financial support and valuable advice; X.S. helped with the reviewing of this article. All authors have read and agreed to the published version of the manuscript.

**Funding:** The authors would like to acknowledge financial support from the National Natural Science Foundation of China [grant no. 51804069, 51804071, and 51974081] and the Fundamental Research Funds for Northeastern University [grant no. N172503015].

**Conflicts of Interest:** The authors declare no conflict of interest.

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
