# Peer review of "The Adsorption Behavior of Moisture on Smelter Grade Alumina during Transportation and Storage—for Primary Aluminum Production"

_metals, doi:10.3390/met10030325_

Round 1

Reviewer 1 Report

This manuscript discusses the process of moisture adsorption on smelter grade alumina, evaluating in particular the kinetic of the process, through a comparison of the experimental data obtained on three samples of alumina, differing for sources and temperature of calcination, with the parameter calculated with a second order kinetic model.

The article is well written and a good characterization of the samples is performed, concerning the surface surface area, the pore size and particles distribution and the water vapour adsorption. However, also a morphological analysis of the samples, optical microscopy or SEM should be considered by the authors for a more complete characterization of the samples.

Concerning the adsorption enthalpy, sample #1 and 3# present an increase with the adsorption temperature in the range 20–60 °C that is not present in the samples 2#. How the authors justify this difference?

As well as the moisture adsorption of the SGA affect the absorbability for HF, that is the major role of the SGA in the dry scrubber, this work does not deal the HF adsorption process. For this reason, the abstract must be modified and set on the topic of the work.

In the opinion of this referee, the manuscript is suitable for the publication on “Metals” after these minor revisions.

Author Response

Reviewer 1

  1. Optical microscopy or SEM should be considered by the authors for a more complete characterization of the samples.

Thank you very much for the reviewing of this manuscript. Yes, the SEM microscopy intuitively shows the particle morphology. However, the reason for not introducing the SEM microscopy in this paper is because the internal microporous structure of the alumina particle dominates the adsorption capacity. The number and shape of these micro pores can be directly described on a pore size distribution curve and indirectly reflected from its specific surface area, while the SEM photos cannot provide quantitative data for analysis.

  1. Concerning the adsorption enthalpy, sample #1 and 3# present an increase with the adsorption temperature in the range 20–60 °C that is not present in the samples 2#. How the authors justify this difference?

  In Table 6, the adsorption enthalpy of sample #2 maintains relatively stable with the adsorption temperature in the range of 20°C-60°C. This is probably because the sample #2 has the smallest average pore size of the three samples, therefore its adsorption capacity is less influenced by the temperature. However, the obtained adsorption enthalpy at the level of 4-13 kJ/mol is sufficient to provide an evidence for the physical adsorption process.

  1. As well as the moisture adsorption of the SGA affect the absorbability for HF, that is the major role of the SGA in the dry scrubber, this work does not deal the HF adsorption process. For this reason, the abstract must be modified and set on the topic of the work.

  The abstract has been modified according to the reviewer’s comment.

Reviewer 2 Report

Please find the pdf with manuscript and comments in the file attached. The introduction is weak and need improvements. You need to define the different types of water in SGA and state which one you are studying in this paper. You also need to define what MOI/LOI is and the temperature ranges that are relevant for these. MOI is not the moisture in the SGA, it is a measuring technique to determine the water content. You also need to clarify what the objective of your work is. Why have you done it and also put it into a greater perspective in the Al reduction cell.

Can you also clarify why you have decided to do all these tests at the various heat treatments?

What does it mean tha you have done tests at 20, 40 and 60 °C? This is not clear to me.

Do not use bullet points in the conclusion.

For the rest of the comments, please see the uploaded manuscript.

Author Response

Reviewer 2:

  1. The introduction is weak and need improvements. You need to define the different types of water in SGA and state which one you are studying in this paper. You also need to define what MOI/LOI is and the temperature ranges that are relevant for these. MOI is not the moisture in the SGA, it is a measuring technique to determine the water content. You also need to clarify what the objective of your work is. Why have you done it and also put it into a greater perspective in the Al reduction cell.

  The definition of MOI and LOI as well as their measuring method have been added into the revised manuscript. The introduction part has been modified accordingly in the manuscript.

  There are two motivations for this work. Firstly, this work introduces the physical adsorption capacity of alumina. Since the adsorption of moisture and HF on SGA are closely related, the investigation on the diffusion rates of gaseous moisture can have some indications on the adsorption behavior of HF on the SGA. Secondly, the SGA adsorbs ambient moisture during storage and transportation, the increased water content in SGA would result in an increase in the cell HF emission after alumina feeding due to electrolyte hydrolysis. This investigation could give some basic data for a better understanding of the cell ‘summer syndrome’.

  1. Can you also clarify why you have decided to do all these tests at the various heat treatments?

  Referring to the heat treatment on the Bay SGA sample #1, the calcination process changed the microstructure of the SGA particles. The dominant phase of SGA changed from γ-Al2O3 to α-Al2O3 after the calcination at 1200 °C for 2h. The phase transformation caused a significant decrease in the specific surface area and pore volume. The measurement data of the adsorption capacity before and after calcination could provide an evidence for the relationship between the moisture adsorption capacity and specific surface area / pore volume.

  1. Do not use bullet points in the conclusion.

  The conclusion part has been modified accordingly.

  1. Can you say something about the motivation for making sample 3?

  China is facing a problem of shortage in high-quality bauxite now. Some alumina companies are trying to find a new method to produce smelter grade alumina from high-alumina-content coal fly ash. Since the Al/Si ratio in the coal fly ash is approximately 1, the Bayer process is not suitable. Some companies have made their first step successfully. The sample #2 tested in this work is a kind of alumina produced from coal fly ash. It is a quite special sample contrast to the Bayer alumina, some comparisons have been made in this work to show some first-hand data.

  Other problems raised in the attached file has been modified in the revised manuscript.

Reviewer 3 Report

Article review

The Adsorption Behavior of Moisture on Smelter Grade Alumina during Transportation and Storage for Primary Aluminium Production.

General appreciation of the article

The article was interesting to read as it is well structured and the results are coherent and clearly explained. This work presents original experimental results on water adsorption on alumina (SGA). The paper presents also a calculation of adsorption enthalpy and a possible adsorption kinetic model of water vapor on SGA.

 Appreciation of the different sections

Introduction is well structured and clearly explains the challenges behind the understanding of the moisture and HF adsorption on SGA. Experimental conditions and characterisation methods used are clearly explained. Characterisation of the alumina is well presented, and each characterisation method has a clearly described purpose and brings new information that the other methods cannot give. 

All the important parts are included: literature review, experimental set-up and procedures and results and discussion. The paper is well written and clear.

Minor corrections to be included:

 p.1 line 17   the sentence is not clear. Some words are missing:  this fact can help reveal the HF…???

p.2 line 51 Experimental Design and Sample…

p.2 table 1 the nature of the alumina calcination process should be included instead of only the origin. I t can help the understand the alumina properties

p.3   figure 1  the line of Sample 3 is not enough visible

p.5 line 139-143 On figure 1, sample 2 , a small fraction of 10-12 micron is present although a narrower particle size range o f this alumina . How this small fraction can affect the adsorption?

Comment for all the paper. This paper shows a lot of experimental results without indication of the errors of the measurements. This information must be provided. After that all the number should adjusted with the correct significant numbers. For example, the number in table 4 present 4 digits after the points, which is not realistic.

p.7 line 202 A explanation of why no distinct pattern is found should be included.

Author Response

Reviewer 3:

  1. 1 line 17   the sentence is not clear. Some words are missing:  this fact can help reveal the HF…???

  It seems something wrong with the reviewer’s computer, anyway, I have checked the revised manuscript to confirm this sentence is complete.

  1. 2 line 51 Experimental Design and Sample…

  Same as comment 1.

  1. 2 table 1 the nature of the alumina calcination process should be included instead of only the origin. It can help the understand the alumina properties.

  Sample #1 was the raw material for the calcination experiments, as re-clarified in Table 1 in the revised manuscript. And the properties of these samples #1, #2, and #3 are compared in detail in Table 2, Fig. 1, and Fig. 2.

  1. 3   figure 1  the line of Sample 3 is not enough visible

  Same as comment 1.

  1. 5 line 139-143 On figure 1, sample 2, a small fraction of 10-12 micron is present although a narrower particle size range of this alumina. How this small fraction can affect the adsorption?

  The authors believe that the micro pores in the alumina particles dominate the adsorption capacity of alumina instead of the fine particles. However, the fine particles of SGA usually contain a larger proportion of alpha phase alumina compared to the bulk alumina. The alpha alumina usually behaves a very small specific surface area and pore volume, which indicates a poor adsorption capacity.

  1. Comment for all the paper. This paper shows a lot of experimental results without indication of the errors of the measurements. This information must be provided. After that all the number should adjusted with the correct significant numbers. For example, the number in table 4 present 4 digits after the points, which is not realistic.

  The measuring techniques (B.E.T. specific surface area, Malvin laser particle size analyzer, Aquadyne DVS Pore size analyzer) employed in this investigation are quite accurate. In this work, the measured data was kept to 4 digits after the point, while most of the calculated values were kept to two digits after the point. The authors suggest to keep these original data since this work provides some first-hand basic data.

  1. 7 line 202 A explanation of why no distinct pattern is found should be included.

  During the phase transformation from gamma alumina to alpha alumina after the calcination experiments, the pore structure of gamma alumina is destroyed. The total pore volume and specific surface area are greatly reduced in alpha alumina. A few micro pores might still exist after the calcination at 1200oC, but the adsorption ability of alumina was almost lost. During the calcination, the pores were destroyed randomly, so the values of the mean pore size and the most probable pore size of the calcined SGA samples in Table 5 showed no obvious principle.

Thank you again for the reviewers’ work.

Round 2

Reviewer 2 Report

I have now looked through the revised paper, and I am a little bit disappointed with the result. The results are interesting enough for publication, but the introduction has to be improved. This is what I need the author to improve in the introduction in order to approve it for publication:

-definition of physisorbed water, chemisorbed water and structural hydroxyl. In this they need to mention the phases gibbsite, boehmite and in which temperature ranges the various waters are released upon heating. This is not correct as it is written in the paper now.

-I mentioned in the first round of review that MOI is a measuring technique to quantify the moisture content in the alumina. It is not correct to write: "increasing the MOI and fluorine content of the secondary SGA." It has to be "increasing the physisorbed moisture and the fluorine..." This needs to be fixed.

-There is little references in the new text author have added to the introduction. This needs to be revised.

-I would also have liked that author clarified the motivation for the various heat treatments and the motivation for making sample 3 in the paper text also, not just in the answers to me.

Author Response

Reviewer 2 (round 2):

  1. definition of physisorbed water, chemisorbed water and structural hydroxyl. In this they need to mention the phases gibbsite, boehmite and in which temperature ranges the various waters are released upon heating. This is not correct as it is written in the paper now.

The introduction part has been enhanced according to the reviewer’s comments, and some conceptual mistakes have been modified in the attached manuscript. The physisorbed moisture and structural hydroxyl in smelter grade alumina is defined and explained with the calcination process of alumina from gibbsite. The phase composition of alumina, including boehmite, gamma, gamma prime, theta, alpha alumina phases are introduced accordingly. The releasing temperatures of boehmite and physisorbed moisture in alumina are presented to explain the measuring methods - MOI and LOI.

  1. I mentioned in the first round of review that MOI is a measuring technique to quantify the moisture content in the alumina. It is not correct to write: "increasing the MOI and fluorine content of the secondary SGA." It has to be "increasing the physisorbed moisture and the fluorine..." This needs to be fixed.

Yes, this is a very accurate description. I have modified in the revised manuscript.

  1. There is little references in the new text author have added to the introduction. This needs to be revised.

Two related references have been added.

  1. I would also have liked that author clarified the motivation for the various heat treatments and the motivation for making sample 3 in the paper text also, not just in the answers to me.

This part has been added to the manuscript.